# A New Design of Poly(N-Isopropylacrylamide) Hydrogels Using Biodegradable Poly(Beta-Aminoester) Crosslinkers as Fertilizer Reservoirs for Agricultural Applications

**DOI:** 10.3390/gels9020127

**Published:** 2023-02-03

**Authors:** Yasemin Balçık Tamer

**Affiliations:** Department of Polymer Material Engineering, Yalova University, 77200 Yalova, Turkey; yasemin.tamer@yalova.edu.tr

**Keywords:** poly(beta-aminoester), poly(N-isopropylacrylamide), crosslinker, swelling, biodegradation, fertilizer

## Abstract

Poly(N-isopropylacrylamide) (P(NIPAAm)) hydrogels were prepared by free-radical polymerization with biodegradable poly (β-amino ester) (PBAE) crosslinkers at 1 wt% and 3 wt% ratio, and compared with conventional N,N′-methylene bisacrylamide (MBA)-crosslinked hydrogel. The influence of the type, molecular weight, and diacrylate/amine ratio of the crosslinker on the crosslink density, compressive strength, and swelling and biodegradation behavior of the hydrogels was investigated. The hydrogels synthesized with lower molecular weight PBAE crosslinkers showed higher crosslinking degrees and compressive strength and lower swelling ratios. To reveal the controlled release behavior of the fertilizer, KNO_3_ was used as the model, and its loading and release behavior from these hydrogels was also examined. The N/T5/1 sample with 1.5/1.0 diacrylate/amine molar ratio and 1 wt% PBAE ratio demonstrated the most controlled release of KNO_3_ with 66.9% after 18 days in soil. In addition, the hydrogel with the porosity of 71.65% and crosslinking degree of 2.85 × 10^−5^ mol cm^−3^ showed a swelling ratio of 69.44 g/g, biodegradation rate of 23.9%, and compressive strength of 1.074 MPa. Thus, it can be concluded that the new designed biodegradable P(NIPAAm) hydrogels can be promising materials as nitrate fertilizer reservoirs and also for controlled fertilizer release in soil media for agricultural applications.

## 1. Introduction

Hydrogels are highly porous crosslinked polymer chains with a three-dimensional network structure that are insoluble in water but can absorb a significant amount of water to swell to several times their original volume due to their hydrophilic functional groups in polymer chains, such as –NH_2_, –COOH, –OH, –CONH_2_, –CONH, and –SO_3_H [1,2,3,4]. The lightly crosslinked structure of hydrogels is produced by chemical or physical crosslinking of hydrophilic polymers through covalent bonds or physical interactions, such as Van der Waals interactions, hydrogen bonding, or ionic interactions [5,6]. Hydrogels have unique properties, such as high swelling ability, porosity, soft structure, flexibility, chemical durability, and biocompatibility, and, thus, to date, they have been used in a variety of applications, such as the food industry, water purification, agriculture, as absorbents and tissue scaffolds, and in drug delivery, pharmaceuticals, personal care products, and in many other fields [6,7,8]. The behavior of a hydrogel can be directly affected by environmental stimuli, such as temperature, pH, light, magnetic field, and electric field, and the hydrogel may swell or shrink in response to these properties. This reversible volume change can be continuous or discontinuous, depending on the chemical nature of the hydrogel [9,10].

Among the stimuli-sensitive hydrogels, poly(N-isopropylacrylamide) (P(NIPAAm)) is a unique thermo-sensitive polymeric network that exhibits a reversible phase transition at around 32 °C, known as the lower critical solution temperature (LCST), in the aqueous medium [1,11]. Furthermore, P(NIPAAm) can undergo conformational changes with ambient temperatures as a result of hydrogen bond formation or breaking of hydrogen bonds between the amide group of NIPAAm chains and water molecules [12]. At low temperatures, phase transition occurs, and P(NIPAAm) chains expand due to the hydrogen bond formation resulting in hydrogel swelling; conversely, phase separation occurs, and the hydrogel shrink due to the breaking of hydrogen bonds and deswelling of swollen hydrogel when the temperature exceeds its LCST value [13,14]. The P(NIPAAm) has both hydrophilic amide (–CONH–) and hydrophobic isopropyl (–CH(CH_3_)_2_–) side groups in its structure which support its thermosensitive property due to the connections between these groups and water molecules [1]. The physical and chemical properties of P(NIPAAm) hydrogels, such as swelling and network structure, change significantly with phase transition. Because of these behaviors, various drug, dye, and fertilizer release systems can be designed that are controlled by external temperature. However, P(NIPAAm)-based hydrogels have some disadvantages, such as poor mechanical strength [15], limited biodegradability [16], insufficient loading capacity [17], and immediate release rates [18,19].

Crosslinking is of great importance to obtain polymeric gel networks that determine the internal structure of hydrogels, and greatly affects the swelling, mechanical properties, and transport of molecules. Although the determination of the properties of a polymeric gel is often controlled by the choice of monomers, which are the main components of polymer gels, the crosslinker used in relatively small amounts compared to the monomer is crucial in designing a hydrogel structure with unique properties. Hydrogels are generally crosslinked with many compounds, such as glutaraldehyde, formaldehyde, dialdehyde, epoxy compounds and various divinyl compounds, such as N,N′-methylenebisacrylamide (MBA) [5]. On the other hand, the use of polymeric crosslinkers in the preparation of hydrogels allows for obtaining amphiphilic networks with unique swelling properties in both water and organic solvents [20]. Until now, PEG macromers have been widely used as a crosslinking agent instead of traditional MBA to prepare P(NIPAAm) hydrogels with improved water retention properties due to their high hydrophilicity and the biocompatibility of PEG chains. For example, Zhang et al. [21] prepared P(NIPAAm) hydrogel microspheres in an aqueous two-phase system using poly(ethylene glycol) diacrylate (PEG-DA) as a crosslinker. More recently, Wu et al. [22] synthesized P(NIPAAm) hydrogels with three different crosslinkers, namely poly(ethylene glycol) diacrylate (PEGDA), glycerol ethoxylate triacrylate (GETA), and citric acid-(PEG acrylate), and investigated the effect of crosslinker type and content on the mechanical properties, swelling behavior, and drug release of the hydrogels, and compared the results obtained with the commercial crosslinker N,N’-methylene bis-acrylamide (NMBA)-based hydrogel. As a result, the development of different divinyl crosslinkers with structural diversity and the determination of their effects on P(NIPAAm) hydrogel properties are a matter of scientific study.

In recent years, poly(beta aminoester) (PBAE) macromers with acrylate end groups have attracted great attention as potential biodegradable and biocompatible crosslinkers that can then be crosslinked using free-radical polymerization. Indeed, PBAE is a kind of polyester that can be obtained from the conjugate addition reaction of primary or secondary amines to diacrylates [23,24]. Since these polymers were first discovered, PBAEs were used in gene and DNA delivery because of the potential for transfection efficiency of their tertiary amine groups [25]. Langer and Lyn [25] have explored a library of PBAE macromers using a single step-growth Michael-type addition polymerization. This synthesis reaction does not contain any additional initiator, terminator, or crosslinker, or any protection system, and no further purification is required after the reaction, as there are no toxic byproducts [26]. The PBAE crosslinkers can be obtained in various structures and in a wide range of molecular weights by altering the type of diacrylate and amine and their molar ratios in which diacrylate is used in molar excess. Due to having hydrolytically cleavable ester groups on their backbones, PBAEs exhibit pH-dependent degradation behavior, and the diacrylate/amine ratio has a direct effect on their degradation rates [27]. Furthermore, the biocompatibility of PBAE macromers has been confirmed both in vivo and in vitro by characterizing non-toxic degradation products, bis(amino acid), diols from the diacrylates, and acrylic kinetic chains [28]. The use of these PBAE crosslinkers can provide outstanding properties to P(NIPAAm) hydrogels, as the swelling and shrinkage behavior of these hydrogels depend on many factors, such as the type and amount of crosslinker, the degree of crosslinking, and the hydrophilic/hydrophobic balance.

On the other hand, these PBAE-crosslinked biodegradable P(NIPAAm) hydrogels have the potential to be used in agricultural applications as they have the ability to retain large amounts of water and offer a controlled release of different forms of fertilizer into soil gradually over time. To meet the growing demand for foods and other agricultural products, over the past few decades, large amounts of fertilizers have been applied to increase crop yields, such as nitrogen, phosphorus, and potassium [29,30]. However, when these fertilizers are applied directly in conventional agricultural applications, the effectiveness of most fertilizers, especially nitrogen forms, is greatly reduced due to the leaching of volatile matter and nutrients. The use of fertilizers can cause environmental pollution and health problems due to the hydrolysis of these nutrients and water eutrophication caused by the effects of microorganisms and dangerous emissions (NH_3_, N_2_O, etc.) [31,32]. In this context, it is important to develop slow-release fertilizers to increase crop yield while reducing nutrient loss to the environment. Since the main purpose of slow-release fertilizers is to provide the nutrients needed by the plant slowly and continuously, slowing the dissolution of fertilizers by applying physical barriers can be an effective approach [32]. In this regard, various polymeric hydrogels, such as poly(lactic acid)/cellulose, polydopamine-graft-poly(N-isopropylacrylamide), and starch-g-poly(acrylic acid-co-acrylamide) hydrogels have been developed to provide a controlled release of fertilizer [29,30,31,32]. However, new hydrogel materials still need to be developed, since the aforementioned hydrogel materials have some drawbacks, such as insufficient mechanical stability, inability to maintain their structural integrity until the fertilizer is released, and not being fully degradable in the soil after the release process.

Although P(NIPAAm) hydrogels have been prepared before, according to the available information, biodegradable PBAE-crosslinked P(NIPAAm) hydrogels have not been prepared so far and have not been used as fertilizer reservoirs for agricultural applications. Hence, it is of great importance to design a well-established environmentally friendly hydrogel that would eliminate the weaknesses of the above-mentioned materials and impart the structure with the qualities of enhanced swelling, mechanical strength, facile degradability, and tunable fertilizer release property. In this hydrogel structure, PBAE crosslinkers undergo degradation through hydrolytically cleavable amino ester groups in its backbone in aqueous environments, leading to disintegration of the structure as it loses its integrity. The degradation rate can be easily tuned over a wide time range by selecting particular amine and diacrylate monomers with different hydrophilic characteristics and adjusting their ratio.

Based on these backgrounds, the aim of this study is to develop a new P(NIPAAM) hydrogel to encapsulate fertilizer and achieve its controlled release, as well as to reveal the influence of crosslinker molecular structure, molecular weight, and ratio on the properties of these hydrogels, especially swelling behavior and biodegradability. First, PBAE crosslinkers were synthesized by Michael-type addition polymerization between two amine compounds of different hydrophilicities and a diacrylate. A series of P(NIPAAm) hydrogels were synthesized by free-radical polymerization using PBAE crosslinkers of different hydrophilicities and the APS/TEMED initiator system. The PBAE-crosslinked hydrogels were evaluated in terms of morphology, swelling, soil biodegradation, and mechanical stability by various characterization techniques. In this study, potassium nitrate was chosen as fertilizer. To reveal the fertilizer release performance, these hydrogels were subjected to nitrate loading, and their release behavior in deionized water and soil was investigated.

## 2. Results and Discussion

### 2.1. Synthesis of (PBAE) Crosslinkers

Biodegradable PBAE crosslinkers were produced by a step growth Michael-type addition polymerization between the diacrylate groups of DEGDA and amine groups of 4-AB or TMDP following a previously reported method [33] (Figure 1). The PBAE crosslinkers were characterized by GPC and FTIR. Since there were no components other than monomers in the reaction medium, no side reaction formation was observed, and the liquid samples obtained were yellow/orange in color. The molar ratios of diacrylate to amine were altered to 1.1:1 and 1.5:1 to obtain high and low molecular weight PBAE crosslinkers, respectively. As shown in Table 1, various PBAE crosslinkers were obtained with molecular weights (Mn) ranging from 1534 g mol^−1^ to 3604 g mol^−1^ and polydispersity (PD) indices ranging from 1.32 to 1.82, relative to polystyrene standards. The GPC traces of PBAE-A5 and PBAE-T5 crosslinkers demonstrated a series of peaks, proving the formation of PBAE oligomers with different molecular weights [26]. In addition, for PBAE-A1 and PBAE-T1 the peaks obtained at lower elution volumes clearly indicated the higher molecular weight of these crosslinkers.

The structures of the PBAE crosslinkers were confirmed by FTIR spectroscopy. The characteristic peaks of 4-AB, TMDP, and PEGDA monomers were detected in the FTIR spectra, as shown in Figure 1. The peaks at 2933 cm^−1^ and 2862 cm^−1^ belong to methyl and methylene vibrations, respectively. The asymmetric vibrations of C–O–C bonds were detected at 1114 cm^−1^. The broad peak at 3449 cm^−1^ corresponds to the stretching of the characteristic hydroxyl peak of the 4-AB monomer in the PBAE-A1 and PBAE-A5 structures [34]. The peaks observed at 1721 cm^−1^ and 1191 cm^−1^ can be assigned to the stretching modes of C=O and C–O of the ester group, respectively. In addition, the stretching peak of the C–N bond at 1239 cm^−1^ and the characteristic absorption peak of the C=C stretching at 1636 cm^−1^ observed in the FTIR spectra showed that poly(β-aminoester) structures with acrylate end groups were obtained. Thus, it can be concluded that PBAE crosslinkers were successfully formed by the conjugate addition of the primary and secondary amines to the acrylate moieties of diethylene glycol diacrylate. Thanks to the unique properties of PBAE, the tertiary amines in the polymer backbones do not react further with diacrylate monomer, thus, preventing polymer crosslinking or branching, leading to the formation of linear polymers [25].

### 2.2. Synthesis of PBAE-Crosslinked P(NIPAAm) Hydrogels

The P(NIPAAm) hydrogels were synthesized via free-radical polymerization using an APS/TEMED redox system to initiate the polymerization of NIPAM in the presence of biodegradable PBAE crosslinking agent. The PBAE, a macromonomer, functions as both a precursor and crosslinker for the formation of P(NIPAAm) hydrogels. Table 2 summarizes the composition of the obtained hydrogels and their experimental results. The FTIR spectra of PBAE-crosslinked P(NIPAAm) hydrogels are given in Figure 2. All spectra exhibited the same absorption peaks, as the components of the hydrogels had similar groups. The spectra show the characteristic peaks of different functional groups of P(NIPAAm) hydrogels, such as C–H symmetric and asymmetric stretching at 2929 cm^−1^ and 2973 cm^−1^, respectively, –CH_2_– and –CH_3_ bending vibrations at 1459 cm^−1^ and 1386 cm^−1^, respectively, C–N stretching at 1368 cm^−1^, N–H bending at 1542 cm^−1^, and C=O stretching of the amide group at 1627 cm^−1^ [35]. In addition, the broad and intense peak observed at approximately 3280 cm^−1^ can be attributed to the N–H stretching of NIPAAm. This peak also proves the formation of a gel structure due to hydrogen bond formation caused by the water of hydration attached to the polymer. Moreover, the disappearance of the C=C double bond stretching vibrations between 1600 and 1670 cm^−1^ in the FTIR spectra of all hydrogel samples confirmed the successful formation of chemical crosslinking between NIPAAm and PBAE [36].

The gel fraction data indicate the measure of removed non-covalently bound reactants from the hydrogel structure during swelling in water. As seen in Figure 3A, the gel fractions were generally quite high, indicating the efficacy of the PBAE crosslinkers in NIPAAm crosslinking. Furthermore, at high crosslinker concentrations (3.0 wt%), lower gel fraction values obtained. The gel fractions of the PBAE crosslinkers were slightly higher at 3.0 wt% crosslinker concentration compared to the MBA-crosslinked N1 and N3 hydrogels.

### 2.3. Morphology and Porosity

The cross-sectional morphologies of the lyophilized P(NIPAAm) hydrogels are displayed in Figure 4. As can be clearly seen from the micrographs, interconnected and relatively homogeneous distributed open pores were obtained. In this study, PBAE-crosslinked P(NIPAAm) hydrogels were prepared by freeze-drying, which removes water by sublimation of ice crystals into water vapor after pre-freezing at −20 °C. With this process, the interconnected and uniform porous microstructure in the hydrogel matrices was achieved with pore sizes between 50 and 300 µm. The N1 and N3 hydrogel samples demonstrated a completely different structure with oval pores when compared to the PBAE-crosslinked P(NIPAAm) hydrogels. In the case of the N/A1/1 and N/A5/1 hydrogels, the largest pores (nearly 250 µm in size) were obtained. It was observed that increasing the diacrylate/amine ratio of the PBAE crosslinker resulted in a reduction in pore sizes, probably due to the increased degree of crosslinking resulting in the formation of a stiffer gel structure. It can be concluded that the crosslinker type did not significantly affect the pore structure of the hydrogel, but that the porosity of lyophilized hydrogel samples gradually decreased with the increase in chemical crosslinking sites in the P(NIPAAm) molecular chains with PBAE crosslinker.

On the other hand, the porosity of the P(NIPAAm) hydrogel samples was determined by the liquid displacement method, in which hexane was used as the displacing solvent that can penetrate into the hydrogel structure without causing swelling or shrinkage. As shown in Figure 3B, P(NIPAAm) hydrogels showed a porosity ranging between 64.57% and 83.12%. It can be concluded that the porosity did not vary significantly between different types of PBAE crosslinkers, but the crosslinker ratio had an effect on the porosity values.

### 2.4. Swelling Studies

The most important feature of hydrogels is their ability to swell easily, which can be determined by the absorption mechanism controlled by the diffusion process. The driving force for swelling are the electrostatic repulsive forces between the network chains and external solvent, as well as the changes in the free energy of the system. Therefore, the nature of the polymeric network and the type and nature of the external solution are of primary importance. This property is influenced by three important factors, namely (i) ionic group content of the hydrogel matrix, leading to electrostatic interaction, (ii) crosslinking density, and (iii) the affinity between the crosslinked hydrogel network and the external solvent defined by the polymer–solvent interaction parameter (χ) [37]. Since P(NIPAAm) is a nonionic polymer and the solvent used is the same, and water is used in all samples, the crosslinking density is of great importance for PBAE-crosslinked hydrogels. The Flory–Rehner theory based on the equilibrium water swelling principle was applied to determine the crosslinking density of P(NIPAAm) hydrogels, and the results are given in Table 2. As expected, the average molecular weight between the crosslinking points decreased with increasing crosslinker ratio, which in turn increased the crosslinking density. Therefore, due to the limitation of water absorption ability at higher crosslinking densities, the maximum swelling percentage decreases with the increasing crosslinker ratio [38,39]. The crosslinking density and, thus, the swelling ratios of P(NIPAAm) hydrogels were evaluated based on three different parameters, namely the structure, chain length, and ratio of the PBAE crosslinker.

Swelling study results for 1.0 wt% and 3.0 wt% crosslinker are given in Figure 5A,B, respectively. From these curves, it is clear that equilibrium swelling was achieved in approximately 6 h and then reached a plateau for all samples. The hydrogels crosslinked with PBAE-A1 and PBAE-T1 crosslinkers resulted in a greater degree of swelling compared to PBAE-A5 and PBAE-T5. In addition, the masses of N/A1-x- and N/T1-x-coded hydrogels doubled in approximately one hour, while it took two hours for N/A5-x and N/T5-x to absorb their own weight in water, regardless of the crosslinker ratio. This result clearly indicated that lower crosslinking density was obtained in hydrogels crosslinked with PBAE, having a diacrylate/amine ratio of 1.1:1. Since this ratio directly affects the molecular weight of the PBAE crosslinker, higher molecular weights obtained in the low diacrylate/amine ratio increase the Mc value between crosslinked chains and cause a decrease in crosslinking density, thus, increasing swelling. Moreover, P(NIPAAm) hydrogels crosslinked with commercial MBA swell less and slightly slower than PBAE-crosslinked hydrogels for both crosslinker ratios, which can be attributed to the large differences in the molecular weight of these crosslinkers and, thus, the higher degree of crosslinking.

As seen in Figure 5A,B, a decrease in the swelling ability of the hydrogels was observed by increasing the crosslinker ratio from 1% to 3% by weight. The maximum swelling degrees of 105 and 79 g/g were reached for N/A1/1 and N/A5/1, respectively, at a ratio of 1 wt%, while drastic decrements in the swelling behavior as 85 and 67 g/g were obtained at a ratio of 3 wt% for N/A1/3 and N/A5/3, respectively. These findings revealed that the crosslinking density of the hydrogels increased as the crosslinker ratio increased, and as a result, the degree of swelling decreased due to the reduction in the spacing between the polymer chains, which led to the formation of a stiffer gel structure that is not capable of absorbing as much water [40]. Thus, the inverse correlation between the swelling rates and the crosslinker ratio was confirmed, which is directly related to the crosslinking degree. In addition, the minimum swelling ratios were observed with N/T1/3 and N/T5/3 due to their higher crosslinker ratio and higher crosslinking densities.

To investigate the effect of the molecular structure of crosslinkers on the swelling behavior of the P(NIPAAm) hydrogels, two different PBAE crosslinkers were employed, namely PBAE-A or PBAE-T. The swelling ratios of the hydrogels increases in the order of N/A1/1 > N/A5/1 > N/T1/1 > N/T5/1, or N/A1/3 > N/A5/3 > N/T1/3 > N/T5/3. This result demonstrates that the swelling ratios of PBAE-A1- or PBAE-A5-containing hydrogels are higher than those of PBAE-T1- or PBAE-T5-containing ones. This difference in swelling behavior of hydrogels is due to the nature of the crosslinkers. Although diacrylate compounds are the same in all PBAE crosslinkers, the structures of the amine compounds are quite different. The 4-AB amine compound has a more hydrophilic structure due to the hydroxyl group in its structure and, thus, it increased the hydrophilicity of the N/A1-x and N/A5-x hydrogel series, resulting in higher SR values. The P(NIPAAm) hydrogels crosslinked with PBAE containing TMDP have a more hydrophobic structure that prevents water molecules from reaching the inside of the network, and lower swelling degrees between 72.5 and 57.2 g/g were obtained. As can be further confirmed from Figure 5, in general, lower crosslinker ratios, the lower diacrylate/amine ratio of PBAE crosslinkers, and the hydrophilic groups incorporated into the P(NIPAAm) hydrogel enhanced the swelling ratio of the hydrogels.

Swelling kinetic analysis confirmed that PBAE-crosslinked P(NIPAAm) hydrogels very closely followed the non-Fickian law of diffusion throughout the entire swelling process. The swelling kinetics of the hydrogels were investigated using the swelling test results only up to 60% of the curves given in Figure 5A,B. The constants *n*, *K*, and *D* were calculated from Equations (7) and (8) and are summarized in Table 3. The diffusion mechanism in polymeric hydrogels can theoretically be described by the constant n value depending on the chemical structure of the hydrogel and the interaction between the polymeric chains and water [41]. For all hydrogel samples, the calculated values of *n* were found be between 0.55 and 0.85, indicating that the transport mechanism of water through P(NIPAAm) hydrogels followed a non-Fickian diffusion process, in accordance with the diffusion mechanism classification proposed by Alfrey et al. [42]. The non-Fickian diffusion process can be classified as Case II diffusion and anomalous diffusion depending on the relative rates of diffusion and chain relaxation. Case II transport takes place when diffusion occurs faster than relaxation, whereas the anomalous transport is identified if the rates of diffusion and relaxation are comparable [43]. The *n* values obtained for P(NIPAAm) hydrogels suggest the anomalous type of swelling where the water diffusion rate is equal to the network chain relaxation. It is clear from Table 3 that the *n* value slightly increased as the crosslinker ratio increased from 1% to 3% by weight. However, in the case of MBA-crosslinked hydrogels, the opposite trend was observed. This result can be attributed to the greater hydrophilicity of the PBAE chains compared to MBA. As the crosslinker ratio increased, a further move away from Fickian diffusion was observed. In addition, higher *n* values were obtained for PBAE-T-crosslinked hydrogels since they have a more hydrophobic structure than PBAE-A-crosslinked gels.

In order to determine the potential of hydrogels in many application areas, analyzing the diffusion of water into the hydrogel structure is of great importance as it characterizes the behavior of the polymer in swelling medium. Accordingly, the diffusion coefficient (*D*) values of the hydrogels were calculated and listed in Table 3. As can be seen, the *D* values of the hydrogels ranged from 2.67 to 4.42 cm^2^ s^−1^, and a slight decrease was observed in the *D* values as the swelling capacity increased. In contrast, hydrogels with lower swelling rates exhibited higher diffusion coefficients. For example, the highest value of *D*, 4.42 cm^2^ s^−1^, was observed for N/3 (SR, 51.4 g/g), and the lowest value of *D*, 2.67 cm^2^ s^−1^, for N/A1/1 (SR, 85.0 g/g).

### 2.5. Mechanical Properties

The mechanical strength of the hydrogels under equilibrium swelling was determined by compressing the hydrogels up to 60% of their initial height using a uniaxial compressive test method, and the results are shown in Figure 6. The commercial MBA-crosslinked P(NIPAAm) hydrogels, N/1 and N/3 demonstrated moderate mechanical properties but brittle behavior with compressive strengths of 0.424 MPa and 0.512 MPa, respectively. As can be expressed in Figure 6, the PBAE-crosslinked hydrogels with more hydrophobic characteristics have a much tougher structure with higher compressive modulus values compared to N/1 and N/3. This enhancement in mechanical strength can be attributed to differences in the internal microstructure of these hydrogels. The higher molecular weight of PBAE crosslinkers compared to low molecular weight MBA helps to disperse the stress exerted on macro-hydrogels by providing enough space for configuration changes in the hydrogel structure to withstand stress even under large strains [44].

Since mechanical tests were performed with samples in the swollen state, the swelling ratio of a hydrogel is of primary importance in determining their mechanical strength. From Figure 6, it is clear that the mechanical performance of P(NIPAAm) hydrogels is highly dependent on their swelling ratio and that hydrogels with high water absorption capacity displayed poor mechanical performance. Hence, the lower the swelling ratio of the hydrogels, the better the mechanical strength. Thus, the mechanical stability of the hydrogels can be evaluated in terms of the ratio of PBAE crosslinkers as well as their molecular structure and diacrylate/amine ratio. In these hydrogel series, the greatest effect on the compressive strength and modulus values was seen with the crosslinker ratio, as the crosslinker ratio of 3 wt% provided the best mechanical performance. The compressive strength for the N/A1 hydrogel sample increased from 0.135 MPa to 0.392 MPa when the PBAE ratio increased from 1% to 3% by weight. The hydrogel structures became more rigid with increasing crosslinker ratio, which directly determines the crosslinking density and, thus, the hydrogel strength.

The analysis results of compressive strength in Figure 6 demonstrated that the change in the diacrylate/amine ratio of PBAE, and, thus, the molecular weight of the PBAE crosslinker, slightly influenced the compressive test results, and compressive strength increased when increasing this ratio from 1.1:1 to 1.5:1. A 44.6% improvement was observed in the compressive strength of N/T5/1, which probably made the crosslinked polymer chains much tighter, resulting in a denser structure compared to N/T1/1. In addition, the type of crosslinker also affected the mechanical properties, so hydrogels crosslinked with highly hydrophilic PBAE/A1 and PBAE/A3 crosslinkers exhibited lower compressive strength due to their high swellability. The hydroxyl functionality of these crosslinkers led to higher swelling of the hydrogel and subsequently affected the mechanical strength of the gel. Thus, the N/A1/1 sample showed a compressive strength of 0.135 MPa when compared to the N/T1/1 hydrogel sample (0.742 MPa). Moreover, N/T5/1 and N/T5/3 hydrogels with low swelling degrees exhibited the highest compressive strength among the entire hydrogel series.

### 2.6. Biodegradation Studies

The biodegradability of the PBAE-crosslinked P(NIPAAm) hydrogels was determined by monitoring the degradation profile of samples buried in soil for 30 days. Biodegradation of hydrogel samples in soil begins with the absorption of soil moisture at the surface and then continues with microbial degradation by interacting with soil microorganisms and enzymes [45]. The percent weight loss of the samples as a function of soil burial degradation time is given in Figure 7. It is clear that all hydrogel samples demonstrated a progressive degradation profile, regardless of crosslinker type and ratio, so their weight and structural integrity were changed due to degradation of the bio-oxidizable components in their structures. As given in Figure 7, the greatest weight loss was observed within 14 days, indicating rapid degradation of the hydrogels due to strong microbial activity, followed by a slower biodegradation until the end of the test, as reported in similar studies [39]. Common P(NIPAAm) hydrogels crosslinked with MBA crosslinker showed the lowest weight reduction, with 10.2% and 5.70% degradation for N/1 and N/3, respectively, over 30 days compared to their counterparts, as the lower molecular weight and hydrophobic nature of the MBA caused the obtained hydrogels to absorb less water, resulting in decreased rates of hydrolysis and degradation. In general, the weight loss decreased as the crosslinker ratio increased from 1 wt% to 3 wt%, so the N/A-x/1 and N/T-x/1 hydrogels degraded faster than the N/A-x/3 and N/T-x/3 hydrogel systems. This is an expected phenomenon because the higher degree of crosslinking achieved in the case of higher crosslinker ratio results in a denser hydrogel structure that prevents the diffusion of water through the hydrogel matrix.

In addition, data from Figure 7 showed that hydrogels crosslinked with PBAE/A crosslinkers were more susceptible to microorganisms in soil, reaching approximately a weight loss percentage of 1.5-fold higher than that of PBAE/T-crosslinked hydrogels in 30 days. Therefore, for hydrogels of similar molecular weights (e.g., N/A1/1 and N/T1/1), the change in amine structure significantly affected the degradation rate of P(NIPAAm) hydrogels, and N/A1/1 (weight loss of 40.9%) demonstrated a faster degradation rate than N/T1/1 (weight loss of 31.8%). This behavior can be explained by differences in the hydrophilic character of these crosslinkers, as demonstrated in their swelling ratio (Figure 5). The degradation of N/A1/1 was found to be significantly rapid, with about 60% weight remaining at the end of the 30-day soil burial test period. Higher hydrophilicity means greater water retention and increases the surface area of the hydrogel for microbial attack, thereby promoting the hydrolysis of ester bonds of the PBAE crosslinkers to yield lower molecular weight degradation products, namely poly (beta-amino acids) and diols, and, thus, the rate of biodegradation is improved [23]. Consequently, the degradation rate of P(NIPAAm) hydrogels could be altered by changing the molecular structure, the diacrylate/ amine ratio of PBAE, and its ratio in the hydrogel preparation.

Moreover, the N/A1/1 hydrogel sample started to lose its integrity after 30 days. A possible explanation for this might be that the gradual degradation of the hydrogels causes the crosslinking density of the hydrogels to decrease, providing a greater chance for microorganisms and enzymes to be taken up and ultimately causing the samples to disintegrate.

### 2.7. KNO_3_ Release Behavior of P(NIPAAm) Hydrogels

Because of their highly porous and permeable nature, freeze-dried hydrogels can absorb water-soluble fertilizers while swelling at room temperature. Evaluating the nitrate loading percentages of the hydrogels given in Table 4, it was observed that although there were no significant differences in the % nitrate loading values of the hydrogels, the nitrate loading percentages of the hydrogels with high swelling degree were slightly higher. Figure 8A,B demonstrate the fertilizer release behavior of PBAE-crosslinked P(NIPAAm) hydrogels at ambient temperature. Among the hydrogels, the highest nitrate release percentages were obtained in the hydrogels crosslinked with a 1 wt% crosslinker ratio, varying between 99.5% and 68.3%, whereas the nitrate release amount of the hydrogels showed quite low values with a 3 wt% crosslinker ratio, which ranged between 56.2% and 29.6%. This can be explained by the lower crosslinking degree and, thus, higher swelling ratios of N/Ax/1- and N/Tx/1-coded hydrogels, which govern the release process of the entrapped ions. In addition, the dense structure of N/Ax/3 and N/Tx/3 hydrogels restrained the penetration of water molecules into the hydrogel and, thus, the release of nitrate ions. As seen in Figure 8A,B, nitrate release ratios of all hydrogels were found to be close to each other, but the maximum release percentage was achieved by the more hydrophilic N/A1/1 with a value of 99.5%. On the other hand, N/1 and N/3 hydrogels exhibited lower nitrate release percentages compared to their counterparts, which can be attributed to their higher degree of crosslinking and lower swelling rates.

However, in agricultural applications, it is expected for a hydrogel to release the fertilizer in a slow manner, while having adequate strength to maintain its integrity and prevent the burst release of fertilizer. In this respect, N/A5/3 and N/T5/1 hydrogels seem to be promising candidates due to their slower fertilizer release rates and improved compressive strength. Figure 8C presents the nitrate release behaviors of N/A5/3 and N/T5/1 hydrogels in soil at ambient temperature. Both hydrogels exhibited an initial burst release with a release amount of 29% and 23%, respectively, followed by a continuously slow release of fertilizer. A final release value of 74% for N/A5/3 and 60% for N/T5/1 was achieved after 18 days. The release behavior of the fertilizer occurs as a result of the absorption and penetration of water through the hydrogel matrix, dissolution of the fertilizer with water, and finally the transport of the fertilizer through the hydrogel to the soil.

## 3. Conclusions

To be used in agricultural applications as fertilizer reservoirs, poly(N-isopropylacrylamide)-based new hydrogels crosslinked with biodegradable poly (β-amino ester) (PBAE) crosslinkers were designed. Biodegradable and biocompatible PBAE crosslinkers were synthesized by one-step conjugate addition reaction between primary or secondary amines and diacrylates at diacrylate/amine molar ratios of 1.1/1.0 and 1.5/1.0. To observe the effect of changes in chemical structure and hydrophilicity of the crosslinker, two different amine compounds were used. By using 1 wt% and 3 wt% ratios of these crosslinkers having different molecular weights and chemical structures, P(NIPAAm) hydrogels were obtained by free-radical polymerization in the presence of an APS/TEMED initiator system. It was observed that the crosslinker type and weight ratio significantly affected the crosslinking density, porosity, compressive strength, and swelling and biodegradation behavior of the hydrogels. The PBAE crosslinker with a more hydrophilic character resulted in higher swelling and degradation rates of hydrogels. On the other hand, a reduction in the molecular weight of the PBAE crosslinker resulted in a stiffer gel structure with lower porosity due to the increased degree of crosslinking. These hydrogels showed controlled release of the fertilizer, KNO_3_, in deionized water and soil, regardless of the type of PBAE. Furthermore, compared with the conventional MBA-crosslinked P(NIPAAm) hydrogels, a significant improvement in swelling ratios, compressive strength, biodegradation rates, and fertilizer release properties were obtained by using PBAE crosslinkers. Having a high potential for use in controlled fertilizer release applications, N/T5/1 hydrogel demonstrated 71.65% porosity, 2.85 × 10^−5^ mol cm^−3^ crosslinking degree, 69.44 g/g swelling ratio, a 23.9% biodegradation rate, 1.074 MPa compressive strength, and 66.9% KNO_3_ release percentage after 18 days in soil.

## 4. Experimental Section

### 4.1. Materials

N-isopropylacrylamide (NIPAAm, ≥99%), diethylene glycol diacrylate (DEGDA, 75%), 4-amino-1-butanol (4-AB, 98%), 4,4-trimethylenedipiperidine (TMDP, 97%), N,N′-methylenebis(acrylamide) (MBA, 99%), ammonium persulfate (APS, 98%), N,N,N,N-tetramethylethylenediamine (TEMED, 99%) and potassium nitrate (KNO_3_, ≥99%) were purchased from Sigma-Aldrich. The used solvents, dichlorometane (DCM, ≥99.8%), diethyl ether (≥99%), hexane (≥95%), and methanol (≥99.8%), were obtained from Merck and VWR Chemicals. All chemicals were used as received without any further purification. Deionized water with a conductivity of 0.072 µS cm^−1^ was used throughout the experiments. The soil used in this study was supplied from a local market.

### 4.2. Synthesis of Poly(beta-aminoester) (PBAE) Crosslinkers

The biodegradable PBAE crosslinkers were prepared by the addition reaction between vinyl groups of the diacrylate (DEGDA) and hydrogen atoms of the amine groups (4-AB or TMDP). Molar ratios of 1.1/1.0 and 1.5/1.0 of diacrylate to amine were used. In a typical experiment, 4-AB (2.43 g, 0.0273 mol) was mixed with DEGDA (6.42 g, 0.03 mol), and the reaction was carried out at 60 °C for 72 h with stirring at 200 rpm. After completion of the reaction, the obtained oil-like yellow liquid macromers were dissolved in DCM and precipitated in the cold diethyl ether to remove unreacted monomers before being dried under vacuum. The resulting crosslinkers were stored at 4 °C. The feed compositions of the PBAE crosslinkers were given in Table 1.

### 4.3. Synthesis of P(NIPAAm) Hydrogels

A series of P(NIPAAm) hydrogels were prepared via free-radical crosslinking polymerization by using various PBAE crosslinkers, including PBAE-A1, PBAE-A5, PBAE-T1, and PBAE-T5 and the APS/TEMED redox initiation system. Hydrogels were synthesized at a crosslinker concentration of 1 wt% and 3 wt% based on the amount of monomer. For both hydrogels, NIPAAM (1.5 g) and a predetermined amount of crosslinker were dissolved in 10.0 mL of deionized water. To this solution, the redox couple of APS and TEMED was added in an amount of 0.2% by weight of the monomer content, and the resulting solution was purged with nitrogen for 30 min to remove dissolved oxygen. This solution was transferred into glass tubes of 1 cm diameters, the tubes were sealed, and then the polymerization was conducted in a water bath at 40 °C for 24 h. After the gelation, the hydrogels were washed with deionized water and methanol to remove unreacted reactants and then freeze-dried. The resulting crosslinked hydrogels were cut into cylindrical shapes of 1 cm length. For comparison, the P(NIPAAm) hydrogels based on a commercial crosslinker, MBA, were also synthesized under the same conditions at a concentration of 1.0 and 3.0 wt%. The feed compositions of various hydrogels were given in Table 2.

### 4.4. Characterizations

Fourier transform infrared (FT-IR) spectra were recorded on a Perkin-Elmer Frontier FTIR spectrometer with attenuated-total reflectance (ATR) apparatus. A total of 32 scans were collected for each sample using a scan range from 4000 to 650 cm^−1^ with a resolution of 4 cm^−1^. The molecular weights of crosslinkers were analyzed by a gel permeation chromatography (GPC) instrument on a Viscotek GPCmax Autosampler system comprising of a Viscotek differential refractive index detector, three ViscoGEL columns (G2000HHR, G3000HHR, and G4000HHR), and a Waters 515 pump. Samples were dissolved in tetrahydrofuran (THF), which was also used as the eluent (flow rate of 1.0 mL min^−1^). Commercial linear polystyrene standards were used to calibrate the system. The cross-sectional pore morphology of freeze-dried hydrogels was examined by scanning electron microscopy (SEM, FEI Inspect S50 system) at an acceleration voltage of 20 kV. To increase conductivity for higher resolution, the samples were gold-coated via a Cressington 108 auto sputter coater.

### 4.5. Determination of Gel Fraction

To determine the gel fraction of the P(NIPAAm) hydrogels, the samples were dried at 40 °C overnight under vacuum and their weights were recorded (*W*_0_). The samples were immersed in deionized water for 24 h at room temperature until equilibrium swelling to leach the unreacted soluble fractions from the gel structure. The samples were vacuum dried again at 40 °C overnight and their final weights were recorded (*W_f_*). The gel fraction was calculated according to the Equation (1) [46]. Five samples were tested for each hydrogel and their average was given.
(1)$$Gel\;fraction\;\left(\%\right)=\frac{W_{0}}{W_{f}}\times 100\;\;$$

### 4.6. Determination of the Crosslinking Density

The crosslinking density of P(NIPAAm) hydrogels was determined according to the equilibrium solvent swelling principle by applying the Flory–Rehner theory [47]. In brief, swelling experiments were carried out by keeping the hydrogel samples of known weight (*m*_0_) in distilled water at room temperature for 72 h. Then, swollen samples were taken out of the water and dried in an oven at 70 °C for 24 h, and their weights were recorded (*m*). The average molecular weight between crosslinks, *Mc* (g mol^−1^), was calculated according to the following Equation (2) [38]:(2)$$M_{c}=-\frac{q_{e}V_{S}\;\left(\phi^{1/3}-\frac{2\phi }{f}\right)}{\ln\left(1-\phi \right)+\phi +\chi \;\phi^{2}}$$
where *q_e_* is the density of the dry hydrogel sample (g cm^−3^), *V_s_* is the molar volume of the water (cm^3^ mol^−1^), *f* is the functionality of the crosslink (*f* = 4 for this system), and *χ* is the dimensionless polymer–solvent interaction parameter of Flory–Huggins (*χ* = 0.502 for P(NIPAAm) and water [48]. Finally, ϕ is the volume fraction of hydrogel in the swollen equilibrium state and can be obtained from the following Equation (3):(3)$$\phi =\frac{(m_{0}-m_{0}f)/\rho_{e}}{(m_{0}-m_{0}f)/\rho_{e}+\left(m-m_{0}\right)/\rho_{s}}$$
where *m*_0_ and *m* are the masses of the unswollen and swollen hydrogel sample (g), respectively, and *ρ_s_
*is the density of water (g cm^−3^).

By using the molecular weight between crosslinks, *Mc*, the crosslinking density (*ν*, mole cm^−3^) of P(NIPAAm) hydrogels can be calculated with Equation (4), as follows:(4)$$\nu =\frac{q_{e}}{M_{c}}$$

### 4.7. Determination of the Porosity of Hydrogels

The porosity of the freeze-dried hydrogel samples was determined by the liquid displacement method [49]. Briefly, samples were soaked in 10 mL of hexane (*V*_1_) for 5 min in a graduated cylinder. The volumes of the hexane and the hexane-swollen hydrogel sample were recorded (*V*_2_). The hexane-swollen sample was then removed, and the remaining hexane volume was recorded (*V*_3_). This measurement was repeated three times for all samples and the average value was calculated. The porosity of the hydrogels was determined according to Equation (5), as follows:(5)$$porosity\;\left(\%\right)=\frac{V_{1}-V_{3}}{V_{2}-V_{3}}\times 100\;\;$$

### 4.8. Swelling Studies

The swelling behavior of the hydrogels were determined gravimetrically by immersing the cylindrical hydrogel samples (1 cm in diameter and 1 cm in length) in deionized water at room temperature. The initial weight (*M_i_*) of the freeze-dried hydrogels was recorded and samples were taken from the solution at predetermined time intervals and slightly dried with a tissue before they were weighed again (*M_t_*). Deionized water was refreshed daily during the swelling process. Three disc specimens were used for each sample, and the swelling rate in percentage terms was calculated according to Equation (6) [22], as follows:(6)$$Swelling\;ratio\;\left(SR,g/g\right)=\frac{M_{t}-M_{i}}{M_{i}}\;$$

The swelling kinetics of hydrogels can be determined by weighing dry and swollen hydrogel samples at specified time intervals. The water diffusion mechanism through the network can be also identified according to Fick’s Law, as shown in the following Equation (7) [2,50]:(7)$$\ln\left(\frac{M_{t}}{M_{eq}}\right)=lnK+n\;x\;lnt$$
where *M_t_* stands for the swelling degree at time *t*, *M_eq_* represents for the equilibrium swelling degree at an infinitely long time, *t* is the time in seconds, *K* is a structural/geometric constant used to identify the swelling ability of the networks, and *n* is the diffusion exponent representing the transport mechanism. This equation can be used for both Fickian and non-Fickian transport mechanisms for hydrogels and is only valid for the initial swelling stages until 60% of the solvent uptake (*M_t_*/*M_eq_*  ≤  0.6). By drawing the plots of ln (*M_t_/M_eq_*) versus ln*t*, the constant *K* and exponent *n* can be determined as the intersection and the slope of the linear lines, respectively. The diffusion exponent, *n*, determines the type of transport mechanism, as follows: (i) *n* = 0.5 for Fickian diffusion, (ii) 0.5 < *n* < 1.0 for non-Fickian diffusion, (iii) *n* ≥ 1.0 for the relaxation-controlled diffusion, and (iv) *n* < 0.5 for less-Fickian diffusion [51,52].

The water absorption mechanism of hydrogel samples can be also explained by the following Equation (8) [53]:(8)$$\frac{M_{t}}{M_{eq\;}}=\frac{4}{L}\;{\left(\frac{D\;t}{\pi \;}\right)}^{1/2}\;\;\;\;\;\;\;\;\;\;\;\;\;\;\;$$
where *D* is the diffusion coefficient and *L* is the thickness of the dried hydrogel sample. This equation can be used for short time absorption behavior of materials, and the average diffusion coefficient values can be calculated from the slope of the graph drawn between *M_t_/M_eq_* and *t*^1/2^.

### 4.9. Mechanical Testing

Before measuring the mechanical properties, the dry hydrogel samples were soaked in distilled water until fully swollen. Uniaxial compressive mechanical strength testing of the cylindrical P(NIPAAm) hydrogel samples (1 cm in diameter and 1 cm in length) was performed using a Zwick/Roell Z1.0 Universal Testing Machine (Zwick GmbH & Co KG, Ulm, Germany) equipped with a 50 N load cell. For each hydrogel sample, five samples were tested with a crosshead speed of 3 mm min^−1^ and compression tested up to 60% of the deformation rate.

### 4.10. Soil Biodegradation Studies

The biodegradability of PBAE-crosslinked P(NIPAAm) hydrogels was analyzed by soil burial test according to a method proposed by Debandi et al. [54]. Briefly, pre-weighed dry hydrogel samples were placed in teabags to protect them from dirt and buried in soil at 5 cm depth. Soil moisture was maintained at 40–50% by sprinkling water every day while also being kept at room temperature. At pre-determined time intervals, hydrogel samples were removed from the soil, washed with distilled water to remove traces of soil from their surface, and dried in an oven at 50 °C until they reached a constant weight. The biodegradability test was performed in triplicate for each film sample. The weight loss of the hydrogel samples was determined by using Equation (9), as follows:(9)$$\%\;\mathrm{weight}\;\mathrm{loss}=\frac{\mathrm{Initial}\;\mathrm{weight}-\mathrm{Final}\;\mathrm{weight}}{\mathrm{Initial}\;\mathrm{weight}}\times 100$$

### 4.11. KNO_3_ Loading and Release Experiments

In order to load a model fertilizer, KNO_3_, into the system, hydrogel samples were immersed in 50 mL of 1 wt% aqueous KNO_3_ solution in order to allow for the penetration of KNO_3_. Then, these hydrogels were removed from the medium, and the remaining concentration of KNO_3_ was determined from conductivity measurements of the aqueous solutions using a Eutech/Oakton PC 510 model conductometer. Three measurements were recorded for each sample and averaged. The loading percentage of KNO_3_ was calculated using the following Equation (10):(10)$${\mathrm{KNO}}_{3}\;\mathrm{loading}\;\left(\%\right)=\frac{c_{0}-c_{d}}{c_{0}\times m_{d}}\times 100\;$$
where the initial and final KNO_3_ concentrations were stated as *c*_0_ and *c_d_*, respectively, and the weight of the dry hydrogel sample was given as *m_d_*.

To reveal the KNO_3_ release experiments, KNO_3_-loaded hydrogel samples were placed in deionized water (100 mL). The conductivity of this solution was measured at predetermined time intervals and KNO_3_ release amounts were calculated using the following Equation (11):(11)$${\mathrm{KNO}}_{3}\;\mathrm{release}\;\left(\%\right)=\frac{c_{t}\times 0.1}{m_{KNO3}\times m_{L}}\times 100\;$$
where *c_t_* stands for the KNO_3_ concentration at predetermined time intervals, *m_KNO3_* represents the amount of KNO_3_ released by the hydrogel, and *m_L_* shows the amount of KNO_3_ loaded into the hydrogel.

The KNO_3_ release experiments were also repeated in soil. In brief, the pre-weighed KNO_3_-loaded hydrogel samples were placed in teabags and buried in the soil. The moisture of the soil medium was maintained at 40–50% by sprinkling water every day, and the soil was kept at room temperature. Samples were removed from the soil every day and soaked in deionized water for 24 h. The KNO_3_ amount in this solution was determined by conductometer and calculated by using Equation (11).

## Data Availability

Not applicable.

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
