# Peer review of "A New Design of Poly(N-Isopropylacrylamide) Hydrogels Using Biodegradable Poly(Beta-Aminoester) Crosslinkers as Fertilizer Reservoirs for Agricultural Applications"

_gels, 2023, doi:10.3390/gels9020127_

Round 1
Reviewer 1 Report
The article titled "A new design of poly(N-Isopropylacrylamide) hydrogels using biodegradable poly(beta-aminoester) crosslinkers as fertilizer reservoirs for agricultural applications" by Y. B. Tamer is very interesting. It discusses the synthesis and characterization of the properties of new P(NIPAAm)-based hydrogels that can be used as carriers for fertilizers.
The author focused mainly on the comparison of oligomeric crosslinking agents with the traditionally used MBA.
The methodology is described correctly. The results are clear and well explained. However, the article needs minor changes to be published in Gels.
Some studies were performed on hydrogel disks (line 260). However, the methodology (lines 159-173) does not describe how these disks were obtained.
Line 227 - It would also be useful to include information on what sizes the swelling test samples were. Also, were they shaped like disks, or was it a powder?
Figure 1. - How author explain the sharp peaks in the GPCs of the samples, which are located most to the right of the head peak (PBAE-A1, PBAE-A5 and PBAE-T5)? What is the molecular weight of the substance present in the samples represented by this peak? The shape of the peak indicates a substance with low pd.
Line – 352-353 – N-H stretching at 3280 cm-1 from PBAE should not be visible as the crosslinking agent content is only 1 or 3%.
Line 416-417 – incorrect Fig. reference. Should be Fig5A and Fig5B.
Line 432 - incorrect Figure reference
Figure 5 - Figure 5 should be in section 3.4 Swelling studies, then it would be easier to read.
Line 457 - incorrect Figure reference
Line 463 - incorrect Figure reference
Based on the high quality of the manuscript I suggest it to be accepted after minor revision.
Author Response
Point 1: Some studies were performed on hydrogel disks (line 260). However, the methodology (lines 159-173) does not describe how these disks were obtained.
Response 1: The methodology for the preparation of cylindrical hydrogel samples were given in “4.3 Synthesis of poly(NIPAAm) hydrogels” part on Page 17. Also, the extra information about cylindrical hydrogel samples was given in “4.9 Mechanical testing” part on Page 19.
Point 2: Line 227 - It would also be useful to include information on what sizes the swelling test samples were. Also, were they shaped like disks, or was it a powder?
Response 2: The information about the shape and dimensions of the swelling test samples was added in “4.8 Swelling studies” part on Page 18.
Point 3: Figure 1. - How author explain the sharp peaks in the GPCs of the samples, which are located most to the right of the head peak (PBAE-A1, PBAE-A5 and PBAE-T5)? What is the molecular weight of the substance present in the samples represented by this peak? The shape of the peak indicates a substance with low pd.
Response 3: I would like to thank the reviewer for this valuable comment. A series of peaks obtained in the GPC traces of the PBAE-A1, PBAE-A5 and PBAE-T5 crosslinkers including the narrow peak located to the right of the head peak can be attributed to the formation of PBAE oligomers with different molecular weights as reported in a previous study conducted by McBath et. al. [1]. In addition, this narrow peak can be explained by the presence of a flowrate marker, a small molecule such as toluene, acetone, or ethylene glycol, which is used to mark the end of the run and can be used to measure column efficiency and identify degradation, tailing, and retention shifts. However, it was deemed appropriate to remove Figure 1 from the manuscript in order to eliminate disagreements and make the article more understandable for the reader. And the molecular weights calculated using the selected baseline and integration limits were given in Table 2.
[1] McBath, R.A.; Shipp, D. Swelling and degradation of hydrogels synthesized with degradable poly(b-amino ester) crosslinkers. Polym. Chem-uk. 2010, 1, 860-865.
Point 4: Line – 352-353 – N-H stretching at 3280 cm-1 from PBAE should not be visible as the crosslinking agent content is only 1 or 3%.
Response 4: The peak at 3280 cm-1 comes from the N-H stretching in the structure of NIPAAM. This sentence was revised to make it more understandable.
Point 5: Line 416-417 – incorrect Fig. reference. Should be Fig5A and Fig5B.
Response 5: The incorrect figure reference referring to the swelling test results has been corrected throughout the manuscript as Fig. 5A and Fig. 5B.
Point 6: Line 432 - incorrect Figure reference.
Response 6: The incorrect figure reference referring to the swelling test results has been corrected throughout the manuscript as Fig. 5A and Fig. 5B.
Point 7: Figure 5 - Figure 5 should be in section 3.4 Swelling studies, then it would be easier to read.
Response 7: Figure 5 was moved to “2.4 Swelling studies” part on Page 9 for the swelling test results to be more understandable.
Point 8: Line 457 - incorrect Figure reference.
Response 8: The incorrect figure reference referring to the swelling test results has been corrected throughout the manuscript as Fig. 5A and Fig. 5B.
Point 9: Line 463 - incorrect Figure reference.
Response 9: The incorrect figure reference referring to the swelling test results has been corrected throughout the manuscript as Fig. 5A and Fig. 5B.
Reviewer 2 Report
This work designed a novel poly(N-isopropy-lacrylamide) composite hydrogel which could be used for agricultural applications as fertilizer reservoirs. Biodegradable and biocompatible PBAE crosslinkers were synthesized by one-step conjugate addition reaction between primary or secondary amines and diacrylates. P (NIPAAm) hydrogels were obtained by free-radical polymerization in the presence of PBAE or APS/TEMED initiation systems. The effects of the crosslinking type, chemical structure, and weight ratio on the crosslinking density, porosity, compression strength, swelling, and biodegradation behavior of the hydrogels were explored. The authors have done a lot of tests, however, there are still some issues needed to be addressed. This manuscript may be accepted after minor modification.
1. “As a result, the development of different divinyl crosslinkers with structural diversity and the determination of their effects on PNIPAM hydrogel properties are a matter of scientific study.” and “However, new hydrogel materials still need to be developed” are mentioned in the article. Could the authors explain the shortcomings of the previous studies to highlight your own research significance?
2. In the mechanical tests, the dry hydrogel samples were soaked in distilled water until fully swollen, while the hydrogels was required to load a model fertilizer KNO3 in agricultural application, whether the two different solutions will make the mechanical properties of the hydrogels different.
3. “Introduction” should be preceded by a supplementary number of 1. The serial number of the three formulas was wrong in 2.10 and 2.11 sections. The data of the gel fraction and porosity in Table 2 should be clearer in the form of bar charts.
4. The font of “The O-H stretching absorption of the hydroxyl group of PBAE-A1 and PBAE-A5 observed as a broad peak at 3449 cm-1.” is not uniform and is repeated with the above. The peak at 1239 cm-1 and 1636 cm-1 in Fig. 2 and the peaks in Fig. 3 should be marked.
5. “Fig. 4” should be “Fig. 5” and the initial mass of the hydrogel should be illustrated when discussing the swelling behavior in 3.4 section.
6. “As it can be expressed in Fig. 6, the PBAE crosslinked hydrogels have a much tougher structure with higher compressive modulus values.” is not an accurate expression. Not all the PBAE crosslinked hydrogels have a much tougher structure , thus “the PBAE crosslinked hydrogels” needing to be indicated clearly.
7. It is necessary to explain which componet of the material to be selected and why for it in 3.7 section.
8. Is it possible to explore the effects of temperature changes on the properties of the hydrogel ?
Author Response
Point 1: “As a result, the development of different divinyl crosslinkers with structural diversity and the determination of their effects on PNIPAM hydrogel properties are a matter of scientific study.” and “However, new hydrogel materials still need to be developed” are mentioned in the article. Could the authors explain the shortcomings of the previous studies to highlight your own research significance?
Response 1: In the “Introduction” part on Page 3, our motivation to carry out this study and the differences of this study from the existing literature were explained by referring the shortcomings of previous studies. In addition, the reasons for the use of PBAE crosslinkers and the advantages of the designed hydrogel structure were given to impart the importance and purpose of the study.
Point 2: In the mechanical tests, the dry hydrogel samples were soaked in distilled water until fully swollen, while the hydrogels was required to load a model fertilizer KNO3 in agricultural application, whether the two different solutions will make the mechanical properties of the hydrogels different.
Response 2: I would like to thank the reviewer for this significant suggestion and awareness. Since hydrogels are highly swollen materials, their swelling ability will directly affect the mechanical properties. As is well known, the swelling behavior of hydrogels is influenced by the presence of counter-ions and hence the ionic strength of the medium. In general, the equilibrium swelling ratio is expected to decrease as the ion concentration increases, due to the reduction of the osmotic pressure difference between the hydrogel and the swelling medium, which weakens the driving force for water diffusion and reduces the equilibrium water content. Thus, the mechanical strength of the hydrogel is afffected. However, this effect was ignored while designing this study, considering that the KNO3 solution used would not have a significant effect on the mechanical strength of the hydrogels due to its low concentration (1% by weight). Nonetheless, this comment has inspired me to take this influence into account when planning my future publications.
Point 3: “Introduction” should be preceded by a supplementary number of 1. The serial number of the three formulas was wrong in 2.10 and 2.11 sections. The data of the gel fraction and porosity in Table 2 should be clearer in the form of bar charts.
Response 3: A supplementary number of 1 was added to the “Introduction” part on Page 1. The serial number of the three formulas has been corrected as Equations 9, 10 and 11 in Sections 4.10 and 4.11 on Page 19 and 20. The results of the gel fraction and porosity analysis were given in the form of bar charts as Figure 3 to make the data clearer.
Point 4: The font of “The O-H stretching absorption of the hydroxyl group of PBAE-A1 and PBAE-A5 observed as a broad peak at 3449 cm-1.” is not uniform and is repeated with the above. The peak at 1239 cm-1 and 1636 cm-1 in Fig. 2 and the peaks in Fig. 3 should be marked.
Response 4: As suggested, the repeated sentence has been removed and the font size adjusted, and also the relevant peaks were marked in Figure 1 and Figure 2 (new graph numbers).
Point 5: “Fig. 4” should be “Fig. 5” and the initial mass of the hydrogel should be illustrated when discussing the swelling behavior in 3.4 section.
Response 5: The incorrect figure reference referring to the swelling test results has been corrected throughout the manuscript as Fig. 5A and Fig. 5B.
Point 6: “As it can be expressed in Fig. 6, the PBAE crosslinked hydrogels have a much tougher structure with higher compressive modulus values.” is not an accurate expression. Not all the PBAE crosslinked hydrogels have a much tougher structure, thus “the PBAE crosslinked hydrogels” needing to be indicated clearly.
Response 6: As suggested, the above sentence has been revised to clarify which PBAE cross-linked hydrogels have a tougher structure as shown in “2.5 Mechanical properties” part on Page 11.
Point 7: It is necessary to explain which component of the material to be selected and why for it in 3.7 section.
Response 7: In the agricultural applications, hydrogels are required to have slow release rates to be more efficient. Considering this factor, N/A5/3 and N/T5/1 hydrogels seem to have optimized composition with slow release rates, moderately good swelling capacity and also good mechanical stability, which is crucial for preventing burst release in these applications. So, as stated in “2.7. KNO3 release behavior of P(NIPAAM) hydrogels” part on Page 15, the degradation test in soil was conducted with only these two hydrogel samples.
Point 8: Is it possible to explore the effects of temperature changes on the properties of the hydrogel?
Response 8: I would like to thank the reviewer for this valuable suggestion. In this study, I focused on the design of environmentally friendly and biodegradable P(NIPAAm) hydrogels crosslinked with different poly(beta-aminoester) crosslinkers to be used as fertilizer reservoirs for agricultural applications, and also characterization of basic properties such as morphology, swelling, soil biodegradation, mechanical stability as well as nitrate loading and release behavior properties, which are of primary importance for agricultural applications. The effect of temperature was not planned to be included in the hydrogel properties. However, I believe that these data would also improve the quality of my paper. This suggestion inspired me to consider temperature effects when planning my future publications.
Reviewer 3 Report
The author has designed new hydrogels. The work is detailed and well presented. There are a few minor details that need to be addressed.
How did the author select 1 and 3 wt.% as the relevant amounts?
In the swelling study, how were "dry hydrogels" prepared?
The font size on page 9, lines 327-329 seems to be too large
Figure 4, SEM images. the scale bar and its text is difficult to read, maybe the author could add a better bar and text
The swelling study refers to Figure 5, yet in this paragraphs the author repeatedly refers to Fig. 4, 4A and B, this needs to be change to Fig. 5, 5A and 5B.
Author Response
Point 1: How did the author select 1 and 3 wt.% as the relevant amounts?
Response 1: The amount of crosslinker used in the hydrogel synthesis directly affects the swelling properties, mechanical strength, degradation rate as well as the release properties of hydrogels. An increase in the crosslinker ratio increases the crosslink density and reduces the swelling and degradation rate due to the increased stiffness and rigidity of the hydrogel and a lower amount of water diffusion. In the case of degradable crosslinkers, an increase in the crosslinker ratio also increases the number of degradable moieties. So, the optimum content of crosslinker is highly desirable to achieve optimum chain mobility and hydration of the macromolecular chain to obtain best properties. Therefore, with reference to previous studies [1,2], the concentration of crosslinker was chosen as 1 wt% and 3 wt% with respect to monomer, NIPAAm.
[1] Chetty, A., Kovács, J., Sulyok, Z., Mészáros, A., Fekete, J., Domján, A., Szilágyi A., Vargha, V. A versatile characterization of poly(N-isopropylacrylamide- co-N,N-methylene-bis-acrylamide) hydrogels for composition, mechanical strength, and rheology. Expr. Polym. Lett. 2013, 7, 95-105.
[2] McBath, R.A.; Shipp, D. Swelling and degradation of hydrogels synthesized with degradable poly(b-amino ester) crosslinkers. Polym. Chem-uk. 2010, 1, 860-865.
Point 2: In the swelling study, how were "dry hydrogels" prepared?
Response 2: The “dry hydrogels” were prepared by freeze drying method. The methodology for the preparation of cylindrical hydrogel samples were also given in “4.3 Synthesis of poly(NIPAAm) hydrogels” part on Page 17. In addition the extra information about cylindrical hydrogel samples was added in “4.8 Swelling studies” part on Page 18.
Point 3: The font size on page 9, lines 327-329 seems to be too large.
Response 3: The font size was adjusted to be the same as the font size of the manuscript on page 9.
Point 4: Figure 4, SEM images. the scale bar and its text is difficult to read, maybe the author could add a better bar and text
Response 4: A new scale bar and text showing magnification were added in to the SEM images in Figure 4 on Page 8.
Point 5: The swelling study refers to Figure 5, yet in these paragraphs the author repeatedly refers to Fig. 4, 4A and B, this needs to be change to Fig. 5, 5A and 5B.
Response 5: The incorrect figure reference referring to the swelling test results has been corrected throughout the manuscript as Fig. 5A and Fig. 5B.